# Control of Cell Identity by the Nuclear Receptor HNF4 in Organ Pathophysiology

**DOI:** 10.3390/cells9102185

**Published:** 2020-09-28

**Authors:** Vanessa Dubois, Bart Staels, Philippe Lefebvre, Michael P. Verzi, Jérôme Eeckhoute

**Affiliations:** 1Clinical and Experimental Endocrinology, Department of Chronic Diseases, Metabolism and Ageing (CHROMETA), KU Leuven, B-3000 Leuven, Belgium; 2U1011-EGID, Institut Pasteur de Lille, CHU Lille, Inserm, Univ. Lille, F-59000 Lille, France; bart.staels@pasteur-lille.fr (B.S.); philippe-claude.lefebvre@inserm.fr (P.L.); 3Department of Genetics, Rutgers University, Piscataway, NJ 08854, USA; verzi@dls.rutgers.edu

**Keywords:** HNF4 nuclear receptor, cell identity, functional genomics, liver and gastrointestinal pathophysiology

## Abstract

Hepatocyte Nuclear Factor 4 (HNF4) is a transcription factor (TF) belonging to the nuclear receptor family whose expression and activities are restricted to a limited number of organs including the liver and gastrointestinal tract. In this review, we present robust evidence pointing to HNF4 as a master regulator of cellular differentiation during development and a safekeeper of acquired cell identity in adult organs. Importantly, we discuss that transient loss of HNF4 may represent a protective mechanism upon acute organ injury, while prolonged impairment of HNF4 activities could contribute to organ dysfunction. In this context, we describe in detail mechanisms involved in the pathophysiological control of cell identity by HNF4, including how HNF4 works as part of cell-specific TF networks and how its expression/activities are disrupted in injured organs.

## 1. Introduction

Multicellular organisms rely on their ability to produce many different cell types with specific phenotypes and functions. Cell-specific activities are implemented through sophisticated regulatory mechanisms, which allow for cell-specific gene expression despite all cells sharing (almost) identical genetic materials. Indeed, differential usage of the genome gives rise to a myriad of unique transcriptomes and their associated cellular phenotypes. This is achieved through the differentiation process when the cellular transcriptome of progenitors is remodeled and specifically oriented towards that of a given mature differentiated cell. Cell-specific transcriptomes define the molecular identity of cells and an understanding of how the latter was established has been extensively studied by developmental biologists. In this context, transcription factors (TFs) with cell-specific or restricted expression have received particular attention. The change in paradigm with regards to the plasticity of differentiated cells and the ensuing growing interest for regenerative medicine [1,2] have been a more recent incentive for identifying key TFs defining cell identity and understanding their mechanisms of action [3,4,5,6,7].

Hepatocyte Nuclear Factor 4 (HNF4), a TF belonging to the nuclear receptor (NR) family with a restricted pattern of expression, has been the continuous scope of studies regarding its role in the control of cell identity in endodermal organs ever since it was identified in the early 90s [8,9]. Here, after a brief introduction to HNF4, we present the current knowledge regarding its mechanisms of action underlying the control of cell-specific functions and discuss how alterations to these activities contribute to the loss of cell identity in organ pathophysiology.

## 2. HNF4 is a Cell-Type Specific TF Required for Cell Identity in Endodermal Organs

### 2.1. HNF4, an Enigmatic NR Whose Activities are Largely Controlled Through Various Isoforms

As a member of the NR family, HNF4 possesses, in addition to a zinc-finger DNA-binding domain (DBD), a well-conserved ligand binding domain (LBD) [8,10]. Following disputed reports that fatty acyl CoA thioesters serve as HNF4 ligands [11,12,13], biochemical studies found that the HNF4 LBD was endogenously occupied by fatty acids [14,15,16,17]. Intriguingly, these studies indicated that fatty acids could be constitutively present within the LBD and, despite a potential role in stabilizing the LBD structure, may not serve as classical exchangeable ligands controlling HNF4 activities [14,15,16,18]. Hence, HNF4 is different from both classical NRs, such as the steroid hormone receptors, and suspected true orphans with no ability to accommodate a ligand in their LBD [10,19]. HNF4 is an ancestral NR [19], and it has been proposed that this particularity is linked to HNF4 phylogeny, and in addition, leaves the debate open regarding the possibility to pharmacologically target this NR [20,21].

In this context, usual mechanisms leveraged by cells to control TF activities may exert a more important role for HNF4 than for classical NRs (Figure 1). The first includes the regulated expression of different isoforms. In addition to HNF4A, mammals have been found to possess another HNF4-encoding gene named HNF4G [22]. HNF4A and HNF4G, which possess conserved DBD and LBD (92 and 79% homology, respectively) but more variable amino- and carboxy-terminal domains [22,23], are together referred to as HNF4 throughout the manuscript. Both are expressed as multiple isoforms with varying transcriptional regulatory potentials [24,25,26]. The diversity of HNF4A-mediated transcriptional regulatory output is further explained by its ability to bind DNA and modulate gene transcription as a dimer [27]. Indeed, in this context, more than 60 potential HNF4A isoform heterodimers with different impacts on gene expression regulation have been documented [24]. While HNF4A and HNF4G most likely form heterodimers, this has not yet been thoroughly investigated [15]. In addition to alternative splicing, *HNF4A* isoforms are generated by two alternative promoters, P1 and P2, giving rise to proteins harboring or lacking an N-terminal transactivation domain [28,29,30,31]. HNF4A isoforms have different abilities to interact with DNA and transcriptional cofactors [24,26,30,32]. Activity of the *HNF4A* gene promoters is developmentally regulated with, for instance, an epigenetically-driven switch from P2 to P1 promoter-derived isoforms during hepatocyte differentiation [33,34]. However, recent work has demonstrated that the P2 promoter was reactivated in the mouse adult liver upon fasting to produce HNF4A isoforms involved in hepatic glucose production, indicating that regulation of alternative *HNF4A* promoter usage remained critical in differentiated cells [35]. In the differentiated compartment of the colonic crypt, HNF4A isoforms primarily stem from the P1 promoter, while P2-driven isoforms are predominant in the proliferative compartment [32]. Additional pathways controlling *HNF4A* expression during development and differentiation include Hippo-Yes1 Associated Transcriptional Regulator (YAP1) [36,37], which has emerged as a central regulator of hepatic differentiation [38,39]. In addition to the different isoforms, HNF4A transcriptional regulatory activities are also controlled by a series of post-translational modifications, which notably allow for regulation of its functions by the cellular metabolic status [40,41,42]. This includes control of HNF4A stability, subcellular localization, and DNA binding by enzymes such as Protein Arginine Methyltransferase 1 (PRMT1), SRC Proto-Oncogene, Non-Receptor Tyrosine Kinase (SRC), Protein Kinase C (PKC), Protein Kinase A (PKA), and AMP-activated protein kinase (AMPK) [40].

While HNF4A has mainly been described as a transcriptional activator promoting recruitment of coactivators [43,44,45,46], reports have also indicated that HNF4A interacted with corepressors [47,48] and may have acted as a direct repressor in some instances [26,49]. This is in line with the context-specific activities of TFs and cofactors, which are increasingly recognized as versatile transcriptional regulators with the potential to both positively or negatively impact on gene expression [50]. Moreover, at given genes, different HNF4A isoforms can alternatively behave as activators or repressors [24]. Indeed, the amino- and carboxy-terminal domains of HNF4A, which differ in both P1 and P2 promoter-driven and alternatively spliced isoforms, modulate its ability to interact with cofactors [30,47,51,52].

### 2.2. HNF4 is Instrumental for Acquisition of Cell Identity in Endodermal Organs

Initially, HNF4A was identified based on its ability to bind regulatory regions of liver-specific genes [53]. In addition to the liver, HNF4A is mainly found within additional endodermal organs including the kidney, the, and the pancreas [8,54]. HNF4G displays a similar restricted expression pattern, except for the liver where low/no expression is observed [25,55]. Many studies, including several using cell-specific HNF4 deletion in mice, have supported the critical role of HNF4 in the functions of the aforementioned organs (reviewed in [54]). This is notably underlined by the role of HNF4 in the acquisition and maintenance of cell differentiation including hepatocytes [56], enterocytes [57,58,59], and renal proximal tubular cells [60]. HNF4 is not required for engagement into these lineages but rather exerts essential roles during later specification stages giving rise to fully differentiated cells [60,61]. On the one hand, HNF4 promotes the expression of a broad set of genes essential for epithelial cell differentiation and cell-specific activities [58,60,61,62,63], while, on the other hand, it inhibits cellular proliferation [59,64,65]. Reminiscent of its role during development, HNF4A is also instrumental for organ regeneration. Indeed, HNF4A was recently shown to support epithelial regeneration in the intestine in response to irradiation in a mouse genetic model [66]. Moreover, HNF4A is required for termination of liver regeneration, which requires inhibition of hepatocyte proliferation and reacquisition of their fully differentiated phenotype [67,68,69]. Accordingly, HNF4A is necessary to maintain hepatocyte identity in adult mice through regulation of liver-specific genes involved in lipid, glucose, and xenobiotic metabolism, as well as blood coagulation [67,70,71]. Further demonstrating its requirement for hepatocyte identity, HNF4A has been identified among the few TFs able to promote in vitro cellular (re)programming towards hepatocytes [37,72,73,74], an observation of high interest when considering the need to develop reliable in vitro models for toxicological studies, disease modeling, and regenerative therapies. Contrary to the liver, deletion of the *Hnf4a* gene alone only triggers a moderate intestinal phenotype [75]. This was recently shown to be due to redundant functions exerted by HNF4A and HNF4G [57]. In this context, deletion of both *Hnf4a* and *Hnf4g* is required to severely impact the intestine through impaired fetal maturation [57,58].

In addition to the aforementioned experimental evidence, the role exerted by HNF4A in metabolic organs in humans has been highlighted by the discovery that mutations within this gene have been linked to various diseases. First of all, *HNF4A* gene mutations have been associated with maturity-onset diabetes of the young (MODY) [76], a monogenic form of diabetes presenting early in life and characterized by deficient glucose-induced insulin secretion by dysfunctional pancreatic β-cells [76]. More than 100 *HNF4A* mutations have been linked to MODY. These mutations lead to HNF4A haploinsufficiency and gave rise to dominant-negative forms in some instances [77,78]. In addition to alterations to pancreatic activities, MODY patients with *HNF4A* mutations also display liver dysfunction as indicated by reduced apolipoprotein levels and altered lipid metabolism [79,80,81]. Interestingly, patient-derived induced pluripotent stem cells bearing an *HNF4A* MODY mutation displayed altered hepatic and pancreatic β-cell identity gene signatures when differentiated in vitro [78]. Mutations in the *HNF4A* gene have additionally or alternatively been associated with renal Fanconi syndrome and inflammatory bowel disease (IBD), attesting to its functions in human renal tubular cells and enterocytes [82,83,84,85]. While loss of HNF4A could precipitate mucosal barrier dysfunction, the exact molecular mechanisms underlining the role of HNF4A in IBD remains to be described [86], but there are strong associations among human genetic *HNF4A* variations and IBD [87].

Altogether, these data indicate that HNF4A is key to the terminal differentiation of several main cell types of endodermal organs including hepatocytes and enterocytes. In this context, how human diseases triggered by *HNF4A* mutations are explained by an impact on acquisition or maintenance of cell identity remains to be more firmly defined.

## 3. Mechanisms of Action in the Control of Cell Identity by HNF4

### 3.1. HNF4 is a Central Node of the Transcriptional Regulatory Networks Controlling Cell Identity

The crucial roles of HNF4 in the control of cell identity and function of endodermal organs are exerted in tight collaboration with additional TFs. HNF4 functionally connects with these TFs at different levels including cross regulation of their gene expression and, as described in greater details in the following section, collaborative binding to and regulation of non-TF cell identity genes important for tissue homeostasis and function [69,88,89]. Indeed, expression of HNF4 is coordinately controlled with that of other cell-specific TFs through extensive auto- and cross-regulatory loops [37,90] that define the typical organization of cell-specific TF networks [6,91]. Analyses of the temporal expression patterns of the main hepatic TFs during liver development, as well as their recruitment to regulatory regions of their own encoding genes, revealed a gradual increase in complexity of the TF networks. Indeed, hepatocyte differentiation is linked to an amplification of the TF cross-regulatory interactions, culminating in dense combinatorial regulatory circuits in mature cells [92]. These multiple regulatory interactions within the hepatic TF networks contribute to the high expression levels of the individual TFs [92]. In line with super-enhancers (i.e., a set of clustered enhancers displaying exceptionally strong indicators of activity such as chromatin opening, active histone marks or TF recruitment) being linked to highly expressed genes required for the establishment and maintenance of cell identity [93], HNF4A was identified as a core TF in super-enhancer-associated networks in human and mouse liver [69,90]. In addition to ensuring high expression of the individual TFs, such gene network organization confers stability of expression, rendering the network more robust against environmental insults which may transiently affect expression or activity of a single TF, as indirectly shown by HNF4A deletion at different stages of hepatocyte differentiation [92]. Hence, involvement of HNF4 in highly interconnected TF networks would contribute to ensure the maintenance of cell identity. While the hierarchy and mutual requirements of the different TFs to establish a cell-specific network is not fully understood, and despite increased robustness provided by dense cross-regulatory loops in differentiated cells, HNF4A strikingly remains essential for expression of other liver TFs in adult hepatocytes [92].

In addition to its key role in TF networks controlling hepatocellular identity, HNF4A is also a central node of transcriptional regulatory networks promoting and stabilizing cell identity in other endodermal organs including the pancreas [94] and the intestine [95] where it acts redundantly with its paralog HNF4G to regulate enterocyte identity [57]. Reminiscent of the observations in liver, HNF4 displays extensive autoregulation and cross regulation with other enterocyte TFs, as exemplified by the reciprocal activation of HNF4 and SMAD Family Member 4 (SMAD4) in murine intestinal villi [58]. Interestingly, implication of HNF4 in TF networks of several cell types is suggestive of a certain degree of redundancy in the control of cell identity. In this context, its level of expression could constitute an additional mechanism in the control of cell fate specification. Indeed, while low levels of HNF4A and Forkhead Box A1 (FOXA1) are a feature of cells bearing the hallmarks of differentiated hepatocytes, high levels of these TFs rather specify an endodermal progenitor state, from which intestinal differentiation can be induced by overexpression of KLF family members [96]. However, HNF4 also controls identity in different cell types by being connected to cell-specific TFs making each identity TF network unique. This is exemplified by the functional interplay between HNF4 and GATA Binding Protein 4 (GATA4) and Caudal Type Homeobox 2 (CDX2) in enterocytes [95,97], One Cut Homeobox 1 (ONECUT1) in hepatocytes [98], or Pancreatic And Duodenal Homeobox 1 (PDX1) in pancreatic cells [99].

In summary, HNF4 functions as part of highly interconnected and self-sustaining TF networks (Figure 1), which increase in complexity throughout cellular differentiation to allow for the establishment and maintenance of cell identity. In the next section, we review the molecular mechanisms by which HNF4A and interconnected TFs further collaborate at the cistrome level to achieve chromatin-based regulation of cell identity gene expression.

### 3.2. Combinatorial Control of Cell Identity at Cis-Regulatory Elements Bound by HNF4 and Collaborating TFs

Numerous studies have demonstrated that HNF4 controls the expression of genes involved in tissue-specific functions of adult endodermal organs including the liver, the pancreas, and the intestine [57,94,100], through widespread binding to their cis-regulatory modules (CRMs). While enhancers traditionally display strong cell-specific activities as compared with promoters [101,102], HNF4A was found to bind not only to liver-specific enhancers and also substantially to gene promoters in adult liver [103]. Indeed, in contrast to TFs such as HNF1A and ONECUT1 whose binding is mostly enhancer-tropic, HNF4A binds directly to the promoters of almost half of the actively transcribed genes in adult liver [94]. Interaction with the TATA-Box Binding Protein Associated Factor 4 (TAF4) subunit of the general transcription factor IID (TFIID), a member of the transcription preinitiation complex, stabilizes HNF4A occupancy of promoters [104] where it facilitates recruitment of RNA Polymerase II (POLR2) [73]. A fraction of HNF4 promoter-bound genes are involved in housekeeping functions, probably more related to the requirement for cell-specific control of cellular homeostasis including HNF4-mediated regulation of cell proliferation in hepatocytes or intestinal cells [57,59,64,65,105]. At liver identity genes, HNF4A recruitment to enhancers, organized into super-enhancers, add up to promoter binding [7]. HNF4A chromatin recruitment to hepatic CRMs is rhythmic over 24 h, likely as a direct consequence of its expression being regulated by the core clock TFs. HNF4A reciprocally modulates tissue-specific circadian gene expression as part of a feedback loop by physically interacting with the core clock TFs [106].

Activities of CRMs are intimately linked to their chromatin structure, which is functionally linked to TF occupancy by a reciprocal influence [107]. HNF4 additionally controls cell identity and function of endodermal organs through tissue-specific modifications of the chromatin structure. In murine hepatocytes, HNF4A is essential for establishing and maintaining active histone and DNA epigenetic states at CRMs through interaction with E1A Binding Protein P300 (EP300) and Tet Methylcytosine Dioxygenase 3 (TET3), respectively [108]. The role of HNF4A in the maintenance of an active chromatin state at liver identity genes is further illustrated by the proposal that it stabilizes histone H3 lysine 4 dimethylation (H3K4me2) at hepatic CRMs [109]. Likewise, HNF4 is also required for chromatin accessibility in the intestine, as demonstrated by the reduced Assay for transposase accessible chromatin (ATAC)-seq signal at CRMs controlling cell identity in the intestinal epithelium of *Hnf4a*/*Hnf4g* double knockout embryos [57]. Interestingly, HNF4 could also be involved in controlling the chromatin folding reported to be necessary for the functional connection between enhancers and promoters [110]. Indeed, HNF4A recruitment to the promoter of HOX Transcript Antisense RNA (HOTAIR), a long non-coding RNA positively correlated with poor prognosis and progression of gastrointestinal cancers, causes the release of a chromatin loop with HOTAIR regulatory elements, hereby exerting an enhancer-blocking activity resulting in HOTAIR gene repression [49].

As already mentioned in the previous section, control of cell identity by HNF4 is exerted in tight collaboration with additional TFs. The aforementioned transcriptional regulatory networks allow for cross regulation of the composing TF’s gene expression and are also characterized by collaborative binding to and regulation of downstream cell identity genes. A large body of evidence has demonstrated binding of tissue-specific CRMs by clusters containing multiple different TFs. Indeed, cistrome profiling of the key hepatic TFs HNF4A, ONECUT1, CCAAT Enhancer Binding Protein Alpha (CEBPA), and FOXA1 revealed that a large majority of their binding sites were shared [100,111]. This observation was confirmed by subsequent studies integrating higher numbers of hepatic TF cistromes which showed that HNF4A bound together with different groups of TFs at distinct CRMs, i.e., promoters or enhancers of housekeeping or liver identity genes [103]. Chromatin recruitment of HNF4A to hepatic CRMs appeared to be highly interdependent, since single nucleotide variants affecting DNA binding of one TF were predicted to concomitantly and preferentially impact recruitment of co-bound TFs [112]. Similarly, co-binding of HNF4A with other intestinal TFs, including SMAD4, occurs at CRMs of enterocyte identity genes [58]. Both tissue-specific and ubiquitous TFs have the potential to collaborate with HNF4A at hepatic CRMs [103]. In general, whether or not ubiquitously expressed TFs should also be considered as drivers of cell identity in addition to cell-specific regulators, remains a matter of debate [7,113]. Noteworthily, while binding sites of individual TFs have shown poor conservation across species [114], combinatorial binding has been found to be evolutionary stable and more strongly associated with liver disease loci identified by genome-wide association studies as compared with binding sites of single TFs [100,111], highlighting the importance of collaborative gene regulation by multiple TFs acting in concert to establish and maintain cell identity. In line with its key role at CRMs characterized by the combinatorial recruitment of multiple TFs, HNF4A binding has recently been shown to be highly enriched at genomic loci with the greatest number of TF co-associations [115]. Combinatorial binding suggests that CRMs serve as platforms where transcriptional regulatory inputs provided by multiple TFs and cofactors are integrated to fine-tune gene expression. As an example, HNF4A, together with Histone Deacetylase 3 (HDAC3) and Prospero Homeobox 1 (PROX1), is part of a transcriptional regulatory module regulating hepatic triglyceride content [89], while it acts in concert with Hes Family BHLH Transcription Factor 6 (HES6) to regulate genes involved in fatty acid uptake and degradation, as well as ketogenesis [88]. Interplay with metabolic sensors such as the peroxisome proliferator-activated receptor alpha (PPARA) and HNF4A PTMs are among mechanisms involved in specifying different HNF4A activities according to the metabolic status [40,41,42,88].

At the molecular level, the detailed mechanisms that allow for integration of multiple TF activities are not firmly established but could include a large array of cooperative or competitive actions with regards to DNA binding, control of chromatin structure, recruitment of cofactors, etc. [116]. In this context, binding and functionality of HNF4 progressively evolves during the course of organ development and maturation, secondary to CRM binding by pioneer TFs, i.e., TFs characterized by the ability to bind nucleosomal DNA in condensed chromatin. For instance, FOXA pioneer factors, which are already expressed in endodermal progenitor cells in the embryo and are essential for specification of the hepatic lineage, and allow subsequent promotion of hepatoblast differentiation by HNF4A [61,117]. A recent study in which *Foxa* genes were ablated in the adult mouse liver demonstrated that FOXA factors were additionally important to maintain HNF4A binding in terminally differentiated hepatocytes [118], contributing to HNF4′s role in cell identity maintenance. In line, during intestinal development, the pioneer-like factor CDX2 induces gut specification [119], while HNF4 factors are required for the next phase which is intestinal maturation through activation of fetal maturation genes [57,95]. Interestingly, the recruitment of non-pioneer factors, such as HNF4, following pioneer factor-mediated specification is not necessarily linked to immediate transcriptional activation but rather pre-marks specific genomic regions for future activation, pointing towards a role for HNF4 as a bookmarking factor [120]. Once recruited to the chromatin at the differentiation stage, the HNF4A cistrome continues to evolve during the process of liver maturation. Indeed, about half of the HNF4A binding sites are uniquely occupied in hepatocytes of either embryonic or adult liver. These differentiation-dependent sites function as temporal enhancers, regulating temporal gene expression patterns [121]. Hence, this enhancer switching enables HNF4A to fulfill distinct roles during the different stages of liver development. Of note, a subset of enhancers is co-bound by YAP1 and its cofactor TEA Domain Transcription Factor 2 (TEAD2) in embryonic but not in adult liver [121], suggesting Hippo signaling influences this enhancer switching.

In addition to being modulated during the course of organ development and maturation, the HNF4 cistrome is also dynamic at adult age. Indeed, the landscape of CRM binding by HNF4 can be reshaped in fully differentiated adult cells to allow adaption to fluctuating environmental conditions. For instance, HNF4A chromatin recruitment is affected by metabolic status, as evidenced by differential hepatic HNF4A cistromes in fasting vs. feeding [88]. Furthermore, chronic consumption of a diet high in saturated fat leads to a 10% increase in HNF4A genome occupancy in hepatocytes, correlated with dynamic alterations in promoter-interacting regions where HNF4A frequently co-binds with CEBPA to regulate gene expression during metabolic adaptation to diet [122]. Likewise, in the intestine, HNF4A also serves as a modulator of the epigenome and transcriptome in diet-induced obesity. In contrast to liver, however, a slight decrease in genome-wide HNF4A occupancy was detected in colonic epithelium of mice fed a high fat diet and HNF4A target genes were enriched among the downregulated genes [123], supportive of tissue-specific mechanisms of HNF4 adaptation to diet. Interestingly, modulation of HNF4 activity by diet may involve the gut microbiome, as evidenced by reduced HNF4A transcriptional activity [123], as well as diminished genome-wide HNF4A and HNF4G occupancy [124] upon microbiota colonization of the murine intestinal epithelium.

In conclusion, the cistrome of HNF4 functionally connects with the cistromes of other tissue-specific TFs to collaboratively orchestrate the expression of genes implicated in cellular housekeeping, circadian rhythmicity, and cell identity (Figure 1). These highly interconnected cistromes are established during organ development but remain able to be modulated in differentiated cells to allow physiological adaptations. However, when the environmental challenges are too strong or too prolonged, profound alterations in HNF4 expression or chromatin binding take place which might precipitate accompanying loss of cell identity and tissue dysfunction, as discussed in the next section.

## 4. HNF4 Loss Contributes to Loss of Identity in Disease

### 4.1. HNF4 Signaling is Transiently Repressed upon Acute Injury as a Protective Mechanism

Modulation of HNF4 signaling through changes in expression or chromatin binding is not restricted to fluctuating physiological circumstances such as those linked to feeding but also occurs in pathophysiological situations. For instance, partial hepatectomy (PHx) of the mouse liver is accompanied by reduced chromatin accessibility as well as H3K27ac levels at HNF4A binding sites, decreased HNF4A genome occupancy, and diminished expression of HNF4A itself [68,69]. In line, reduced HNF4A expression and binding were identified as major contributors to the suppression of hepatocyte-specific genes and downregulation of liver biosynthetic functions during the liver repopulation process after acute injury in the *Fumarylacetoacetate hydrolase* (*Fah*) null mouse model [125]. Decreased HNF4A expression, subsequently leading to the collapse of the TF networks controlling identity, has been observed in additional murine models of acute liver damage including sepsis and drug-induced acute liver injury resulting from the administration of a single high dose of acetaminophen or carbon tetrachloride (CCl4) [69,90]. HNF4A loss upon acute injury also occurs in the intestine, with both HNF4A expression and chromatin binding being suppressed in a mouse model of acute colitis [84,126]. Similarly, HNF4A recruitment at enhancers and super-enhancers of key renal epithelial genes was reduced in the ischemia reperfusion model of acute kidney injury [127]. In addition to aforementioned evidences from murine models, repression of HNF4A upon acute injury has also been observed in humans, as evidenced by the decreased expression of HNF4A and other members of the TF networks controlling hepatocellular identity in livers from patients with acute acetaminophen intoxication or hepatitis B virus-induced acute liver failure [90].

Pathological loss of HNF4 expression/activity can be triggered by inflammatory signals such as tumor necrosis factor (TNF) through inhibition of *HNF4* encoding gene promoters or of HNF4 transactivation domains by nuclear factor κB (NF-κB) [128,129]. Additionally, endoplasmic reticulum stress (ERS) leads to reduced HNF4 recruitment to the *Cebpa* and *Pparg coactivator 1 alpha* (*Ppargc1a*/*Pgc1a*) promoters [130], in line with ERS repressing genes involved in hepatic metabolic functions including genes regulating lipid homeostasis [131,132]. This is linked to SRC kinase-induced HNF4A proteasomal degradation upon acute ERS and subsequent loss of *HNF4A* mRNA expression [68,69,133]. Importantly, in line with HNF4 controlling hepatic cell identity, our group has redefined the paradigm related to acute ERS-induced changes in the liver by pointing to profound transcriptomic alterations linked to a global loss of hepatic molecular identity [69]. Impaired identity gene expression is characteristic of acute liver injury and is linked to partial hepatic dedifferentiation. Indeed, the transcriptome of injured livers resembles more that of newborn livers than that of mature adult livers [69]. In accordance, a dedifferentiation process accompanies loss of HNF4A activities upon acute kidney injury [127], and intestinal enterocytes may undergo transdifferentiation to goblet cells upon combined loss of HNF4A and HNF4G [58], suggesting that HNF4 plays a similar role preserving cell identity across these tissues.

Since inflammation and ERS are intimately linked [134], both may combine to trigger loss of HNF4 and associated TF networks controlling cell identity in acute organ injuries. Importantly, the extent of the inhibition is related to the severity of the insult (i.e., a strong insult results in a more profound downregulation as compared with a milder perturbation) (Figure 2), as illustrated by the dose-dependent decrease in *HNF4A* transcript levels in murine hepatocytes subjected to ERS [69]. In addition, numerous observations support the notion that this repression upon acute injury is transient. Indeed, when ERS resolves, hepatic HNF4A is re-expressed [69]. In line, after the rapid decline in nuclear and cytoplasmic HNF4 protein levels observed 1 h post PHx, HNF4A returns to baseline levels after three days [68]. The same holds true for re-expression of the liver identity TF network, including HNF4A, after initial suppression in acute sepsis [69]. Temporary shutdown of cell identity, when defined as a transient decrease in the expression of genes involved in tissue-specific functions such as HNF4 factors, may be beneficial in situations of acute injury to permit the cells to reallocate their transcriptional resources. Indeed, competition for transcriptional supplies has been proposed to rule transient transcriptional adaptations to environmental disturbances in a model referred to as transcriptional ecosystem [135]. Hence, the transient decrease in the expression HNF4 and other TFs involved in cell identity upon acute injury may be necessary to allow the increased expression of stress/injury handling genes [69], as well as genes involved in DNA synthesis and cell proliferation enabling repopulation of the damaged liver [68,125] (Figure 2). Recent work demonstrating negative regulation of HNF4A by cyclin D1 post PHx, in which basal hepatic metabolic function is broadly diminished by the cell cycle machinery to allow for cellular resources to be redirected to the demands of growth and proliferation, supports this model [65]. A similar proposal has been made with regards to ischemic acute kidney injury where dedifferentiation would be necessary for surviving kidney cells to proliferate [127]. Hence, the temporary shutdown of cell identity upon an acute insult could be considered to be a mechanism of ”physiological health”, a recent concept evoking the active processes of protective adaptations enabling maintainence of health in a diseased state [136]. Defining stress-induced loss of cell identity as a protective mechanism would reveal the logic of observations such as those describing HNF4 as required for a proper ERS response, through the regulation of the *CrebH* and *Xbp1* basal gene expression [137,138].

### 4.2. HNF4 Signaling is Persistently Repressed Upon Chronic Injury Contributing to Organ Dysfunction

While the initial loss of cell identity gene expression may be beneficial to cope with an acute insult, detrimental effects occur if the identity program cannot be re-established. Indeed, hepatocyte-specific deletion of HNF4A in mice resulted in 100% mortality post PHx [68]. Mechanistically, this effect could partly be attributed to a failure to re-express other identity TFs involved in termination of liver regeneration [69], indicating that acute injury decreases the hepatic TF network robustness. This is most probably linked to the global loss of identity TF expression disrupting the cross-regulatory loops, therefore, rendering hepatocytes more sensitive to loss of activities of individual TFs [139,140]. In line, unmitigated ERS impedes re-establishment of the hepatocellular identity transcriptional program [69] and leads to aggravated hepatosteatosis, due to the persistent repression of hepatic genes involved in fatty acid oxidation [132]. In line with loss of identity TF expression being instrumental in triggering liver injury, forced hepatic expression of HNF4A or FOXA2 has been shown to be protective [68,141,142], while the identity TF levels in livers from deceased septic humans correlated with the extent of liver dysfunction [69]. HNF4A knockout in mouse intestinal epithelium hampered the formation of regenerative foci following lethal irradiation, in line with HNF4A being required for intestinal regeneration [66]. Consistently, HNF4 factors in the intestine were observed to promote lipid oxidation and renewal of intestinal stem cells [105]. Altogether, re-expression of HNF4 and associated identity TF networks after initial decrease is crucial for re-establishment of identity gene expression, and hence restoration of organ function. In this context, persistent downregulation of HNF4 is often encountered in chronically diseased liver and gastrointestinal organs. Indeed, HNF4A is significantly decreased in intestinal tissues from Crohn’s disease and ulcerative colitis (UC) patients [126], while HNF4G is diminished in patients with active UC as compared with UC patients in remission and healthy controls [143]. Reduced hepatic HNF4A levels were found in a mouse model of alcoholic steatohepatitis (ASH), as well as in patients with ASH or cirrhosis [90,144]. Patients with non-alcoholic steatohepatitis (NASH) have also been characterized by low hepatic HNF4A expression, in line with the dramatically decreased HNF4A levels observed in livers from mice with genetic or diet-induced obesity [145]. In accordance, repeated CCl4 administration in rodents, a model of liver fibrosis, represses HNF4A expression [90,142]. Downregulation of HNF4A in fibrotic livers is partly mediated through the Rho/Rho-associated protein kinase pathway [146] and associated with a collapse of the TF networks controlling hepatocellular identity, as illustrated by the repression of hundreds of liver-specific genes in mice fed a methionine- and choline-deficient high fat diet [69]. Hence, HNF4 loss might contribute to diminished cell identity in chronic intestinal and liver injuries [147], which may precipitate accompanying organ dysfunctions. Indeed, when analyzing HNF4A expression in liver tissue from patients at different stages of decompensated liver function, a correlation was found between HNF4A levels and the extent of liver dysfunction, stage of fibrosis, and serum levels of total bilirubin [148]. Interestingly, HNF4 expression also decreases with age, as illustrated by the lower HNF4A transcript levels in pancreatic islets from aged rats as compared with younger animals [149]. Hence, it is tempting to speculate that the age-related decline in HNF4 levels might contribute to the increased incidence of tissue dysfunction with aging, including a higher prevalence of type 2 diabetes (T2D) and of its complications in the elderly [150] (Figure 2).

### 4.3. Deregulated HNF4 Expression and Activities in Cancer

Chronic liver diseases including hepatic fibrosis and cirrhosis may evolve to hepatocellular carcinoma (HCC). In line with the HNF4A loss observed in chronic liver injuries, a number of studies have reported decreased HNF4A expression in HCC. Indeed, lower HNF4A transcript levels were found in rodent models of hepatocarcinogenesis, as well as in HCC patients [151,152,153], although not all studies concur [154]. Consistent with HNF4A being a liver tumor suppressor, deletion of *Hnf4a* promotes the occurrence of HCC in mice [155,156]. In line, repressed HNF4A signaling has been shown to underly the tumorigenic effects of isocitrate dehydrogenase mutations [157]. Conversely, enforced expression of HNF4A blocked HCC occurrence in a rat model [152]. Reduced HNF4A expression might contribute to loss of hepatocellular identity and HCC progression through different mechanisms, among which alterations in epithelial-to-mesenchymal transition (EMT) regulation. EMT is an essential process during tissue development and repair, but it can adversely cause organ fibrosis and promote carcinoma progression [158]. A body of evidence has demonstrated that HNF4A inhibits EMT, in part through direct repression of master regulatory EMT genes such as *Snail Family Transcriptional Repressor 1* (*Snai1*/*Snail*) and *Snail Family Transcriptional Repressor 2* (*Snai2*/*Slug)* [152,159,160]. Recently, modulation of chromatin topology has been implicated in the repression of EMT genes by HNF4A [49]. Furthermore, HNF4A also indirectly inhibits EMT through induction of microRNAs targeting EMT genes [161], inhibition of beta-catenin signaling which promotes EMT [152], and positive modulation of miR-29 which, in turn, limits DNA Methyltransferase 3 (DNMT3) levels, a DNA methyltransferase with a pivotal role in the epigenetic control of EMT [162]. Kinetic analyses revealed that EMT exhibited three consecutive waves of gene expression, and HNF4A was found to particularly regulate the intermediate so-called partial EMT state [163]. Additional mechanisms contributing to HNF4A-related HCC progression include increased cell proliferation [67,164] and reduced apoptosis due to decreased expression of Apoptosis signal-regulated kinase 1 (ASK1/MAP3K5), an apoptosis-promoting tumor suppressor gene which was recently identified as an HNF4A target in hepatocytes [165]. Recent work has revealed that HNF4A-mediated repression of hepatocyte proliferation was based, at least in part, on its inhibition of cyclin D1 [65]. Hepatocellular oncogenesis is further sustained through an HNF4A-microRNA inflammatory feedback circuit, comprising a fading of HNF4A-driven Signal Transducer and Activator of Transcription 3 (STAT3) inhibition and maintenance of HNF4A suppression [166]. In summary, the abovementioned findings point towards a reduced HNF4A signaling in HCC and demonstrate a key role for HNF4A in HCC progression through modulation of EMT, cell proliferation, apoptosis, and inflammation. In line with HNF4A’s role in controlling identity of additional endodermal organs, diminished HNF4A expression is likewise found in renal cancers and cholangiocarcinoma [153]. Interestingly, the molecular mechanisms underlying the persistent downregulation of HNF4A in chronic liver injuries and HCC may at least partly differ from the repressive mechanisms triggered by acute tissue damage described in the previous section. These mechanisms include transcriptional repression by WT1 (Wilms tumor 1), absent in healthy liver but highly expressed in fibrotic and cirrhotic livers [167], and by P53 which is upregulated in various conditions associated with chronic stress [168]. Activation of mitogen-activated protein kinase signaling in HCC reduces HNF4A expression by preventing the recruitment of the basal transcriptional machinery to the proximal promoter, as well as blocking the binding of several TFs including HNF1A and ONECUT1 to enhancers controlling *HNF4A* gene transcription [169]. Signaling induced by transforming growth factor beta (TGF-β) can also dampen HNF4A activities by altering its post-translational modification pattern, including glycogen synthase kinase (GSK)-3β mediated phosphorylation, leading to reduced HNF4A DNA binding and targeting to the proteasome [170,171,172]. Loss of HNF4A expression in a chronic setting can further stem from induction of the oncoprotein gankyrin which enhances proteasome-dependent HNF4A degradation in hepatoma cells [173] and the persistent YAP1 activation in HCC [174]. Indeed, YAP1 negatively regulates HNF4A levels through proteasomal degradation [151], while HNF4A reciprocally inhibits YAP1 expression through direct recruitment to its promoter and additionally represses transcriptional activity of the YAP1-TEAD complex through a competitive mechanism [151,175]. In line, YAP1 inhibition restored hepatocyte differentiation in advanced HCC, partly through reactivation of HNF4A signaling [174]. Some studies have indicated that while P1-driven HNF4A isoforms were lost in HCC, fetal P2-driven HNF4A isoforms coulc be re-expressed [176]. In human colon tumors, P1-driven HNF4A is either lost or localized in the cytoplasm due to an isoform-specific phosphorylation of HNF4A by SRC tyrosine kinase affecting protein stability, nuclear localization, and transactivation function of P1-driven, but not P2-driven, isoforms [133]. In contrast, P2-driven HNF4A is induced in colon cancer though combined actions of Paired Box 6 (PAX6) and HNF1A [177]. In line with the P1:P2 ratio playing a key role in the pathophysiology of colon cancer, tumor load and size were increased in mice which could only produce P2-driven HNF4A isoforms in a model of colitis-associated colon cancer, while restricted expression of P1-driven isoforms and led to fewer and smaller tumors [32]. Accordingly, P1-derived HNF4A acts as a tumor suppressor in a colon cancer xenograft model, while P2-derived HNF4A does not [178]. Altered the P1:P2 expression is additionally found in gastric carcinoma [179]. Note that chronic diseases of endodermal organs also show dysregulation of HNF4A isoform expression with re-expression of fetal P2-driven HNF4A in livers of ASH and T2D patients [35,180]. Further illustrating that cancer relies on processes involved during development, the reciprocal downregulation between HNF4A and SNAIL is at play both in liver stem cells and in HCC [160,175].

In contrast to HCC, where an overall diminished HNF4A signaling is observed in most studies, as discussed above, several gastrointestinal adenocarcinomas (GIAC) including esophageal, gastric, pancreatic, and colorectal adenocarcinomas are characterized by increased HNF4A expression [181,182]. This finding was recently confirmed by a large-scale transcriptomic and DNA methylomic study on 907 GIAC samples from The Cancer Genome Atlas [153]. The latter work furthermore revealed that HNF4A overexpression in GIAC was driven by both genomic and epigenomic mechanisms, as illustrated by the high prevalence of *HNF4A* gene locus amplification in GIAC and the increased chromatin accessibility of its promoter, respectively. In line with its expression being increased in GIAC, inhibition of HNF4A signaling suppressed the growth of GIAC cancer cells and xenografts [153,183]. Conversely, overexpression of HNF4A in adult mouse esophageal explants induced a columnar-like expression profile [184] and HNF4A alone was sufficient to drive chromatin opening and activation of a esophageal adenocarcinoma-like chromatin signature when expressed in normal human epithelial cells [185], suggestive of a key role for HNF4A in promoting GIAC initiation and progression. Hence, while the tumor suppressive functions of HNF4A are evident in liver and are in accordance with its role in the maintenance of hepatocellular identity and quiescence, the increased HNF4A expression and tumor promoting actions in GIAC might suggest that, in certain types of cancer, the typical activities of HNF4A as gatekeeper of cell identity are re-rooted towards other functions which, on the contrary, support tumorigenesis. In contrast to the numerous studies investigating HNF4A’s role in cancer, little is known regarding HNF4G. A recent study however revealed an intriguing role for HNF4G in castration-resistant prostate cancer. Indeed, HNF4G was found to act as a pioneer factor for the aberrant gastrointestinal lineage transcriptional program found in one-third of these patients, while ectopic HNF4G expression in a prostate cancer cell line reduced its sensitivity to hormone deprivation [186].

## 5. Conclusions

HNF4 is undoubtedly a critical regulator of cell identity in endodermal organs. Beyond its role in defining molecular identity at the transcriptomic level during cellular differentiation, control of HNF4 expression is used in adults to allow cells to handle various stresses through partial dedifferentiation and subsequently recover by re-establishing their original identity. Alterations to this process are observed in chronic diseases where HNF4 activities are compromised. Hence, restoration of cell identity through (isoform-specific) modulation of HNF4 expression and/or activities may offer potential routes for therapeutic intervention. However, we would need to deepen our understanding of the intimate control of HNF4 functions, including the specific roles of HNF4A and HNF4G isoforms, as well as a more thorough understanding of their functions in cell-specific TF networks. In this context, a definite answer to the long-lasting question of the opportunity to target HNF4 with synthetic ligands would be instrumental.

## Figures and Tables

**Figure 1 cells-09-02185-f001:**
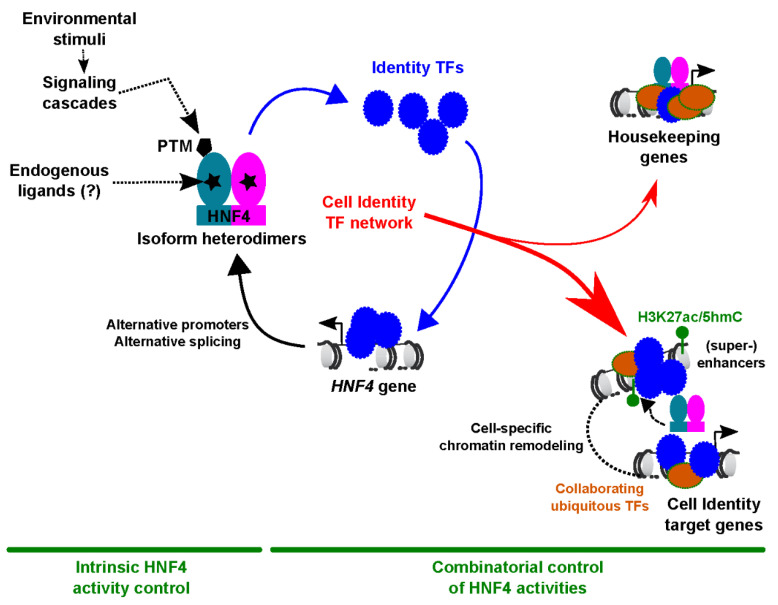
Mechanisms involved in the control of cell identity by HNF4. Intrinsic control of HNF4 activities through isoform heterodimers, post-translational modifications (PTMs), and potential ligands add up to the modulation of its functions through involvement in cell-specific transcription factor (TF) networks. Combinatorial control of cell identity by HNF4 and additional cell identity by TFs consist of 2 interdependent layers where these TFs modulate each other’s expression and coordinately regulate common (non-TF) cell identity target genes.

**Figure 2 cells-09-02185-f002:**
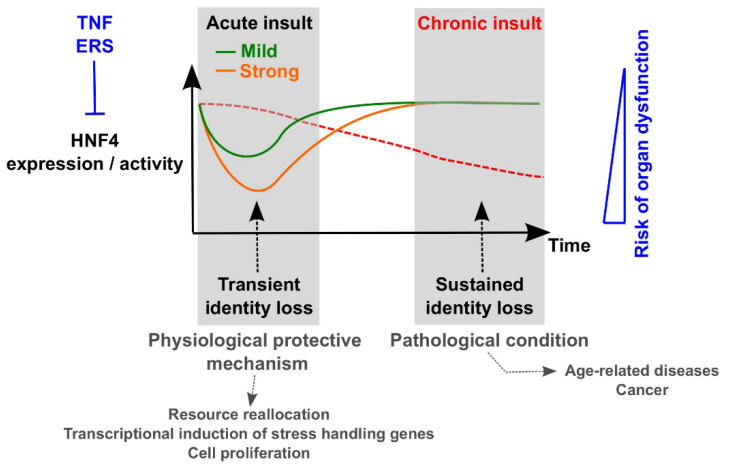
Proposed relationship between strength and duration of organ insults and HNF4-mediated control of cell identity. Acute injury triggers loss of HNF4 transcriptional activity through decreased expression and/or chromatin recruitment, with the degree of the inhibition correlating with the severity of the insult. HNF4 loss is accompanied by a shutdown of cell identity, partly attributable to a collapse of the TF networks controlling identity gene expression. This loss is transient (i.e., expression of HNF4 and other identity TFs is re-established once the stress resolves) and seems to be a protective mechanism to cope with the acute insult in the context of reallocation of transcriptional resources. However, when the insult persists leading to chronic stress/injury, repression of HNF4 and cell identity endures, which might precipitate accompanying organ dysfunction and occurrence of cancer.

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
