# Peer review of "Control of Cell Identity by the Nuclear Receptor HNF4 in Organ Pathophysiology"

_cells, 2020, doi:10.3390/cells9102185_

Round 1

Reviewer 1 Report

This manuscript is a thorough review of the nuclear receptor HNF4, giving emphasis in its role in cellular differentiation as well as in the expression of cell specific genes. it is well written and covers adequately all current literature.

Author Response

We thank the reviewer for his/her positive evaluation of our manuscript.

Reviewer 2 Report

The authors present a comprehensive review article on the role of HNF4 protein in the establishment and in the maintenance of cell identity in endodermal organs. In particular, they collected literature data about i) the molecular mechanisms involved in the control of HNF4 activity, ii) the HNF4-driven transcriptional regulatory networks controlling cell identity and iii) the contribution of the HNF4 loss to endodermal organ dysfunction and disease.

The issue is interesting and the manuscript is well-organized. Interestingly, the specific mechanisms regulating HNF4 activation and function are described in the context of general regulatory mechanisms and in light of the more recent findings.

As suggestion, the authors could consider to mention, among the mechanisms underlying the persistent downregulation of HNF4 in HCC, the HNF4 transcriptional inactivation and posttranslational modifications by TGFbeta (a cytokine playing a key role in EMT and in the molecular pathogenesis of HCC) and its transcriptional repression by Snail. Besides, the HNF4 and Snail reciprocal downregulation represents a regulatory circuit whose balance can influence the maintenance of hepatocyte and liver stem cell identity and differentiation, and the EMT/MET dynamics.

Author Response

As suggested by the reviewer, we now refer to TGFbeta-mediated control of HNF4A activities in HCC (lines 444-447). Moreover, we have included a sentence with regards to the control of hepatocyte stemness by the HNF4/SNAIL reciprocal downregulation (lines 467-468).

Reviewer 3 Report

General comments: In this manuscript, the authors have comprehensively reviewed role of HNF4alpha in hepatic differentiation and in injury/disease at molecular level. The review is timely and broad with exception of some specific areas as described below. The main issue is lack of a coherent theme from beginning to end, which has resulted information overload without synthesis of a central theme. May be the authors can choose a theme such as from development to end stage diseases or differential function between cell types/ organs or normal vs. disease etc. Detailed comments: 1. HNF4a isoforms have distinct functions, which are very important for differentiation of different organs. This should be elaborated in the review in its own specific section. 2. Figure one looks amateur. Should be redrawn with the help of professional artists. 3. There are some functional differences in HNF4a between hepatocytes, kidney cells (proximal tubular epithelium), pancreas and the gut. This should be addressed together in a separate section. 4. Certain important references on liver cancer work are missing. For example, Walesky et al. Hepatology 2013

Author Response

Our review covers the literature related to HNF4 activities in the control of cell identity and a discussion of how alteration to these activities relates to acute or chronic organ insults. The focus of this review is stated both in the abstract (lines 14-16) and at the end of the introduction (lines 41-44). A detailed discussion of organ-specific functions of HNF4A is therefore not the central goal of our manuscript. As now indicated on line 100, this has been previously reviewed in Ref 54 and our manuscript focuses more specifically on the role of HNF4 at the cellular level. In this context, and in line with the reviewer’s comment, we have added specific examples of TFs collaborating with HNF4 in cell-specific networks (lines 178-181).

Our manuscript recognizes the importance of HNF4A isoforms, whose description composes the majority of section 2.1. Accordingly, and in line with the reviewer’s recommendation, we have modified the title of this section to make it clearer that the importance of isoforms is being discussed. This title now reads “HNF4, an enigmatic NR whose activities are largely controlled through various isoforms” (line 46). Moreover, isoforms are also further discussed in the section related to modulation of HNF4 activities in cancer (lines 454-466).

Finally, several references related to liver cancer have been added to the manuscript. This includes studies describing that Hnf4a deletion promotes HCC (lines 409-410), TGFbeta controls HNF4A activities in HCC (lines 444-447) and repressed HNF4A signaling underlies the tumorigenic effects of specific mutations associated with liver cancer (lines 410-411).

We appreciate that professional artists would be able to improve the graphical aspect of Fig.1 but we nevertheless feel that our figure is of enough quality to convey its message.

Reviewer 4 Report

This manuscript by Dubois et al. provides with an extensive review of the recent literature related to the role of HNF4 transcription factors during liver and gut physiology.  My only minor comments are to integrate and discuss one additional reference to better support their conclusions:

1- Line 109: They noted that HNF4 displays a critical role in maintenance of cell differentiation in enterocytes. One reference (59, Cattin et al.) states that HNF4A is crucial for this matter. However, it is clear that deletion of HNF4A alone does not result in a strong phenotype as demonstrated in Babeu et al., 2009.  This is the basis for recent papers where deletion of both HNF4A and G supported redundancy between these factors.  The authors should better discuss this aspect.

Author Response

In response to the reviewer’s suggestion, we now more explicitly state that deletion of both Hnf4a and Hnf4g is required to have a strong intestinal phenotype in the mouse (lines 116-119). Note that the sentence the reviewer referred to indicates that HNF4 (hence both A and G) is instrumental for enterocyte differentiation and cites several references including those related to their redundant functions in these cells.